# Damages induced by the 25 April 2015 Nepal earthquake in the Tibetan border region of China and increased post-seismic hazards

Zhonghai Wu [a*] Guanghao Ha[a], Patrick J. Barosh [b], Xin Yao [a], Yongqiang Xu [c] and Jie Liu [d]

a Institute of Geomechanics, Chinese Academy of Geological Sciences, Beijing 100081, China
b P.J. Barosh and Associates, 103 Aaron Avenue, Bristol, RI 02809, USA and Visiting Research Fellow, Chinese Academy of Geological Sciences, Beijing 100081 China
c China Institute of Geo-environment Monitoring, Beijing 100081, China
d College of Resource Environment and Tourism, Capital Normal University, Beijing 100048, China

**Abstract:** The seismic effects in Nyalam, Gyirong, Tingri and Dinggye counties along the southern border of Tibet were investigated during 2-8 May, 2015, a week after the great Nepal earthquake along the Main Himalaya Thrust. The intensity was VIII in the region and reached IX at two towns on the Nepal border; resulting in the destruction of 2,700 buildings, seriously damaging over 40,000 others, while killing 27 people and injuring 856 in this sparsely populated region. The main geologic effects in this steep rugged region are collapses, landslides, rockfalls, and ground fissures; many of which are reactivations of older land slips. These did great damage to the buildings, roads and bridges in the region. Most of the effects are along four incised valleys which are controlled by N-trending rifts and contain rivers that pass through the Himalaya Mountains and flow into Nepal; at least two of the larger aftershocks occurred along the normal faults. And, the damages are not related to the faulting of N-trending rifts but distributed along the intensity of Nepal earthquake. Areas weakened by the earthquake pose post-seismic hazards. Another main characteristic of damages is the recurrence of the old landslide and rockfalls. In addition, there is an increased seismic hazard along active N-trending grabens in southern Tibet due to the shift in stress resulting from the thrust movement that caused the Nepal earthquake. NW trending right-lateral strike-slip faults also may be susceptible to movement. The results of the findings are incorporated in some principle recommendations for the repair and reconstruction after the earthquake.

**Key Words:** Nepal earthquake, Himalaya Mountains, Seismic hazard, Post-seismic hazardp

---

Corresponding author: Wu Zhong-hai, E-mail: wuzhonghai@geomech.ac.cn,

40

## 1. Introduction

On 25 April 2015 at 14:11:26 MGT+8 (Beijing Time), a great Ms 8.1 (Mw 7.8) earthquake struck Nepal and adjacent regions killing more than 8,800 people and injuring more than 23,000. The epicenter was near Pokhara 77 km northwest of the capital of Kathmandu and the hypocenter was at a depth of 10-24 km. Many aftershocks of magnitude 4.5 $M_w$ or greater followed, of which a Ms 7.5 (Mw 7.3) aftershock occurred after 17 days, on 12 May 2015 at 15:05. This epicenter was near the Chinese border 77 km east-northeast of Kathmandu and the hypocenter was at a depth of 12-16 km.

The main earthquake occurred on the south slope of the Himalaya Mountains and formed a 120-140 km long, about 80 km wide rupture zone with a dip-slip of 3.5-5.5 m, which shows an expansion from west to east (USGS, 2015 a, b; IRIS, 2015). The aftershock distribution, the focal mechanism solution and the source rupture inversion suggest that the earthquake was a release of built-up strain along the Main Himalaya Thrust fault zone and part of the ongoing process of the Indian Plate underthrusting the Eurasian Plate (Fig. 1). This was the strongest seismic event since the 2005 Ms 7.8 Pakistan Kashmir earthquake, which also occurred along the Main Himalaya Thrust. These earthquakes may indicate that the seismic activity along the thrust is entering a new active phase.

The earthquake affected Nepal, northern India, Pakistan, Bhutan, and the southern Tibet of China. Main damages characteristics have been reported in Nepal (Bijukchhen et al., 2017; Yun et al., 2015). However, there is lack of damages investigation in China, which is the focus of this paper. In China the tremors were felt in Xigaz ê and Lhasa to the north and over 300,000 $km^2$, but were strongest in the China-Nepal border area which is only about 40 km (Fig. 1, 2) from the epicenter (Table 1). Despite the great loss of life in Nepal the disaster caused only 27 deaths, 856 injuries and 3 missing in China, although the damage was extensive. About 30 thousand people were affected

and the direct economic loss is more than 33,000 million Yuan (RMB) (5.178 trillion U.S. dollars). Fortunately, the border area has a low population density and the earthquake occurred in the afternoon when many were outside, otherwise the casualty and economic loss would have been much higher. Due to the rapid response of the local governments the displaced people were soon resettled in southern Tibet.

An emergency seismic hazard investigation group of 12 people was organized by the Ministry of Land and Resources to survey the hardest hit four counties of Nyalam, Gyirong, Tingri and Dinggye during 2-8 May, a week after the main shock, in order to quickly understand the earthquake effects and potential future threats to provide a basis for the post-earthquake reconstruction. The group then held meetings with the local governments to present their findings and recommendations. This paper is a brief summary of the direct effects observed in the field and investigations into the delayed effects that may cause as much damage.

## 2. Seismic-Geological Setting

The Tibetan Plateau is well known for its numerous E-W to NW, north-dipping thrust faults (MHT) that facilitated its rise as the India plate collided and was thrust beneath it. Most of the uplift occurred by the Miocene (Dewey et al, 1988; Wu et al., 2008) and the majority of the thrust faults came to a stop as the movement evolved and concentrated along fewer strike-slip faults, which remain very active and capable of great earthquakes (Armijo et al., 1989; Fig. 1). However, thrusting remains dominant in the collision zone at the south edge of Tibet south of the Himalaya Mountains with the continued northward movement of India. Here the greatest activity occurs along the very shallow north-dipping Main Himalaya Thrust, which gave rise to the Nepal earthquake and has a long history of great earthquakes along its length (Fig. 1). Less generally known are a series of nearly N-S-trending normal faults and grabens to the north of the great thrust that complement some of the movement across it. These also are capable of producing significant earthquakes, although they are much shorter in length (Wu et al, 2011). This array of active faults plus a set of NW right-lateral

strike-slip faults, which may aid extension, constitute the seismic framework of the
region.
The China-Nepal border region is located on the south slope of the Himalaya
Mountains close to the Main Himalaya Thrust and contains many active normal faults
that control the transverse valleys that lead into Nepal. The high, rugged, steep
topography and the well-developed incised river valleys in this region further amplify
the destruction caused by earthquakes. Therefore, it is not strange that southern Tibet
was greatly affected by the Nepal earthquake.
**3. Methods and data**
The intensity was evaluated using the Chinese seismic intensity scale (GB/T
17742-2008) (CSIS), which is a revised national standard implemented in 2009, that
has 12 degrees of intensity (GB/T, 2008). This was modified from the GEOFIAN
(Medvedev) scale, that in turn was adapted from the Modified Mercalli scale and is
closely aligned with it, except in the lower units (Barosh, 1969) and is approximately
the same in the higher units reported on herein. The latest CSIS scale revision added an
additional building type for evaluation in reflecting newer construction in the country.
A broad region of southern Tibet was affected by the earthquake, but the sparse
population and difficult terrain did not permit defining of the felt area well. Most
isoseismics for the lower intensities were compiled by the China Earthquake
Administration, which made a quick, overall survey of towns in order to assess the
damage (Fig. 1). However, a detailed field survey, reported below, was made in the
region most affected. The principal effects of the earthquakes are the damage suffered
by structures, highways and bridges and the landslides, collapses and rockfalls. The
landslips caused much of the damage to the construction. Overall 2,699 houses and one
temple were destroyed, 39,981 houses and 242 temples seriously damaged, and about
2,600 km of main highways, 263 bridges, and a part of the communication, power and
water facilities were damaged to some degree in southern Tibet as reported by the
China Earthquake Administration. In the region more closely studied in the field the
damage and seismic intensity were evaluated at 29 sites in 10 affected counties (Fig. 2,
and Table 1).
**4. Results**
**4.1. Damage features and seismic intensity**
Landslides, rockfalls and collapses are common widespread occurrences during
large earthquakes in the mountainous regions of Tibet. The Nepal earthquake was no
exception, even though there was no nearby surface fault offset. The 2008 Ms 8.0
Wenchuan earthquake and its aftershocks at the eastern edge of Tibet produced
hundreds of thousands of such landslips (Wang and Han, 2010; Tang et al., 2011; Yang
et al., 2015). They caused major destruction and casualties, in addition to blocking river
valleys and forming reservoirs that threatened downstream communities. It was only a
massive emergency effort by the government that prevented additional major
calamities. Several small dams were formed by the Nepal earthquake, but no large ones
that needed an emergency excavation, although the threat remains.
The perception of the earthquake, damage to buildings of different material and
structure, and surface effects show obvious differences as recorded at the different
levels of intensity. Only a few people in a room might have felt the earthquake in Lhasa
at intensity III, whereas, to most of the people both inside and outside of buildings in
Xigazê city the earthquake was obvious and demonstrates an intensity IV and strong
damage indicates approximately intensity IX at the Nepal border. The increasing and
varying degrees of damage of buildings and disruptions of the surface in the VI to IX
intensity zones were reviewed in the field in southern Tibet nearer Nepal (Table 1). The
intensity described herein is a composite of both the main shock and the large after
shock. This may have caused an enhancement of the ratings if they were for the main
shock alone, because some structures weakened by it were further damaged or
destroyed by the large aftershock.
Of the four counties investigated, Nyalam County is located on the south slope of
the Himalaya Mountains, whereas Gyirong, Tingri, and Dinggye Counties are located
north of the mountains (for their seismic intensities, see Table 1). The main effects and
economic losses are concentrated in Nyalam, Tingri, and Gyirong Counties (Fig. 2)
where about 80% of the houses were completely destroyed or damaged to a large extent
(Figs. 3, 4). The damage is heaviest in the towns of Zhangmu in Nyalam County; Jilong
and Sale in Gyirong County, and Rongxia in Gyirong County (Fig. 5). Moreover, the
highways and communications to the towns of Zhangmu, Tingri, and Resuo Bridge as
well as connections to Zhangmu, Tingri, Chentang and others in Nyalam County were
greatly damaged and broken.

The Chinese intensity scale considers the varying effects on different building

types and this usually improves the reliability of the general intensity assignment, but
locally it may lead to assigning different values, if there is a greater variation in damage
than usual between types. This could be the case in these areas where the effects appear
to reach either intensity VIII or IX depending on the type of structure used to assign
intensity. The apparent highest intensity, IX, from destruction, that equaled some parts
of Kathmandu, for older self-built stone masonry or adobe structures with poor seismic
resistance, whereas for the newly built cement-bonded stone, brick or concrete
structures it was no more than intensity VIII and the rating lies between (Figs. 3 and 4).
For example, in Jifu Village about 2.4 km south of Jilong, all the houses built of stone
block masonry were almost completely destroyed, whereas most newly built ones of
cement-bonded stone or brick are still standing with only minor cracks in the walls (Fig.
4c-d), and the same variation also is seen at the Sale Town Primary School (Fig. 4e).
The inhabitants of this area had to be quickly moved to temporary settlements (Fig. 4b).
Perhaps some poorer buildings weakened by the first earthquake were collapsed by the
second one or the newer buildings had more seismic resistance than realized. Some
undetected ground slippage at a few locations throughout the region also may have
augmented the effects to a slight degree.

The E-W elongation of the intensity pattern (Fig. 2, Table 1) shows at least twice

the rate of attenuation northward towards the Himalaya Mountains than in an E or W
direction. This can be attributed to the absorption of the seismic energy by the
E-W-trending fault structure and lithologic units of the great Himalaya Mountain block,
plus a contribution from the E-W spread of the earthquakes and aftershocks.
The geologic effects caused by the Nepal earthquake are mainly landslides, terrace
and loose material collapses and debris flows, rockfalls, and ground fissures that were
studied in detail at 33 sites in four towns in Nyalam, Gyirong, Tingri and Dinggye
Counties (Figs. 2, 6). These vary with the intensity, amount of rock weakened by
previous movement, steepness of slope and lithology. These landslips diminish in
number and size northward from the Nepalese border with the decrease in intensity. In
the areas approaching intensity IX landslides and collapses are widespread and include
some large landslides; in the area encompassing intensity VIII small collapses and
landslides were common, but large landslides are rare; intensity VII areas contain some
small landslides, collapses and rockfalls along valley slopes and roadcuts; whereas in
the area of intensity VI small collapses and landslides are rare and a small amount of
rockfalls occurred near roadcuts.
These damages have the following characteristics:
(1) They are all disrupted slides, as classified by Varnes (1978; updated by Hungr
et al., 2014), with a loss of internal cohesion.
(2) They occur most densely along four incised river valleys, which are controlled
by N-S-trending rifts that pass through the Himalaya Mountains and enter into Nepal
(Fig. 2). The four valleys, successively from west to east, are: the Gyirong Zangbo
valley that follows the Gyirong Graben and extends southwards (Figs. 5b and 6e), the
Boqu River valley that follows the Nyalam Graben and passes through Zhangmu and
connects to the Sunkoxi River valley in Nepal (Figs. 5a and 6a); the Rongxiaqu valley
that follows the southwest side of the Kong Co-Gangga Graben to pass through
Rongxia and descend to the Sunkoxi River valley in Nepal (Fig. 5c) and, the Pengqu
River valley, controlled by the Paiku Co Rift, that crosses the Kung Co-Gangga Graben
and the Pengqu Graben southwards and passes through Chentang to connect  to the
Arun River in Nepal (Fig. 5d). The topographic relief in these valleys is generally about
2,000-3,000 m, which is very favorable for various landslips during seismic events.
Furthermore, there is an overall tendency for the number and size of collapses,
landslides, and rockfalls to increase towards Nepal along these valleys.
Remotely-sensed images issued by Google Earth after the earthquake show that the
Gyirong Zangbo and the Buqu River valleys contain the maximum density and scale of
collapses and landslides (Figs. 5a, 5b, and 6a-6h).
Moreover, some dammed lakes due to the collapsed rock and soil can be seen in
these valleys of Nepal. For example, in the Gyirong Zangbo valley, a 0.07 km$^2$ dammed
lake and a 0.04 km$^2$ dammed lake occur about 2.5 km north of and about 7.3 km
southwest of Dhunche Village, respectively, and in the Boqu River valley, a 0.24 km$^2$
dammed lake occurs on the north side of Dabi Village.
(3) Damages occur often in weak, soft geologic material and unstable geomorphic
positions: joint or fault-formed, high, steep bedrock cliffs and slopes (Figs. 6b and 6e);
high, steep slopes of loose Quaternary sediment forming river terraces, proluvial fans,
and benches (Figs. 6d and 6f) and; unstable slopes and highway road cuts (Figs. 6g and
6h).  These landslides mostly occur on slopes steeper than 35-45 degrees.
(4) Most of large ground fissures are associated with collapses and landslides. They
either occur on the displaced masses or around their edges and only a few such fissures
occur on surface of loose sediments (Fig. 8).
These rock and soil slips caused the most serious casualties and damage. The worst
collapse found occurred in Disigang Village about 0.8 km southwest of Zhangmu
where about 0.016 km$^3$ of debris destroyed four or five buildings and killed seven
people (Figs. 4a, 6b and 6c). The largest landslide found occurred about 1.3 km
southwest of Chongse Village near Jilong where about 2,700,000 m$^3$ of material
blocked the main highway from Jilong to Gyirong Port (Fig. 6e). In addition, 27 small
landslides and collapses occurred along the 14 km length of highway stretching from
this landslide to Gyirong Port.
**4.2 Recurrence of seismo-geological hazards**
An important discovery was that the earthquake induced landslide and collapse
generally occurred where previous ones had taken place and correlated in size with the
previous ones. This apparently reflects the effects of ancient earthquakes and provides
new evidence for paleo-seismicity both in location and size. More significantly this
demonstrates the areas of ancient landslide and collapse indicate the potential areas of
danger from further landslips from torrential rains and future earthquakes; important
considerations in seismic risk evaluation and the post-earthquake reconstruction
process.
The collapses and landslides commonly result from reactivation of older ones and
similar effects produced by historic earthquakes occurred near the same position as in
this earthquake. Such features are notable on both banks of the Boqu River near
Zhangmu (Figs. 7a and 7b). At Disigang Village of Zhangmu, for example, a house
built on the side of a large rock brought down previously was destroyed by a new large
rockfall (Fig. 6c). This is a warning that reconstruction after the earthquake, not only
should avoid as far as possible potential new hazards, but at the same time be aware of
previous ones and make a comprehensive assessment of their stability.
The specific structural damage is usually related to the material and type of
building construction in these areas where the heaviest destruction occurred near the
Nepal border. In this region of few trees most of the houses are of simple stone and
adobe construction and these fared poorly during the earthquake; with the majority
being destroyed near Nepal (Figs. 3b, 3d, 3e, 4c). Houses with cement bonded stone or
brick construction survived much better (Figs. 3a, 4d) and those of good brick or
reinforced concrete construction suffered the least (Figs. 3c, 4f) and provided a contrast
with those of poorer materials (Figs. 3a, 4e).
**4.3 Post-seismic Increased Potential Geological Hazard**
The Nepal earthquake has left many potential dangers in its wake in this region and
nearby seismically active areas in southern Tibet that do not fit into an intensity scale
yet are a consequence of the earthquake and pose a serious hazard that might create
even greater damage and casualties than the immediate effects. The delayed effects in
southern Tibet are the consequences of earthquake-loosened landslides and weakened
rock that may fall due to aftershocks and torrential storms and from secondary
earthquakes due to changes in the stress field resulting from the Nepal earthquakes.
Rock, terrace material and previous landslides loosened by the earthquake, but still
in place may fail with small aftershocks and torrential rains, which further weaken the
material and add weight.  Such an increase in secondary landslips during rainy seasons
following earthquakes has been noted previously and is becoming a recognized hazard.
Increased rainfall-triggered landslide activity above normal rates have been observed
after several large earthquakes; two of which occurred in similar terrane to the east and
west (Hovius et al., 2011; Saba et al., 2010; Tang et al., 2011; Dadson et al., 2004). Rain
may even be a factor during an earthquake. Data in New Zealand suggest that
earthquakes that occur during wetter months trigger more landslides than those during
drier periods; although a clear relationship between rainfall-induced pore pressure and
earthquake-induced landslide triggering has not been shown (Dellow and Hancox,
2006; Parker et al., 2015). When the typhoon Toraji hit Taiwan following the 2005 Mw
7.6 Chi-Chi earthquake, 30,000 more landslides occurred with many being reactivated
ones triggered by the earthquake, although 80% of the Toraji landslides occurred in
areas that had not failed during the earthquake (Dadson et al., 2004). The proportion of
surface area disturbed by the landslides increased towards the active fault suggesting
that even in areas that underwent no landslips during the earthquake the substrate was
preconditioned to fail through loss of cohesion and frictional strength of hill slope rock
mass caused by the strong seismic motion (Dadson et al., 2004).
A similar general weakening of rock strength was deduced in the mountainous
region of South Island, New Zealand where to test the possible influence of previous
earthquakes in preconditioning the ground for landsliding, the areas of overlap of high
intensity of similar strong, >Mw 7, 1929 and 1968 earthquakes were compared (Parker
et al., 2015). Many landslides produced in 1968 were reactivations or enlargements of
ones that failed in 1929, but others were not, although there was a higher degree of
failure in the overlapped area than could be easily explained in considering all the
factors normally involved in landslip. These observations suggested that hill slopes
may retain damage from past earthquakes, which makes them more susceptible to
failure in future triggering events, and this had influenced the behavior of the landscape
in the 1968 earthquake. It was further suggested that the damage legacy of large
earthquakes may persist in parts of the landscape for much longer than the observed
less than 10 year periods of post-seismic landslide activity and sediment evacuation.
Similarly, data from the 2010 Mw 7.1 Canterbury – 2011 Ms 6.3 Christchurch
earthquake sequence reveal landslide triggering at lower ground accelerations
following the February 2011 earthquake, which caused cracks to develop in hill slopes
that subsequently failed in later earthquakes in the sequence (Massey et al., 2014a,
2014b; Parker et al., 2015).
This general loosening of rock by ground motion also occurred at the Nevada Test
Site where deep underground nuclear explosions, similar to shallow earthquakes,
caused widespread movement along joints within the overlying bedrock (Barosh, 1968).
Some fractures were propagated upward through 610 m of alluvium to demonstrate
how even this soft material was weakened further.
The progressive brittle damage accumulation in hill slope materials may lead to
permanent slope displacement that results in cracking and dilation of the mass, which
makes them more susceptible to failure (Petley et al., 2005; Nara et al., 2011; Bagde
and Petroš, 2009; Li et al., 1992b; Parker et al., 2015). Whether or not a hill slope fails
in response to an earthquake thus, becomes a function of both a current event and the
history of damage accumulated from previous events (Parker et al., 2015).
These later landslides pose all of the same dangers as those occurring during the
initial earthquake and also may cause damming of rivers to create dangerous reservoirs
that can fail with devastating effects.
The landslips also increase the hazard from flooding in the disturbed region. They
contribute debris to valleys to widen them and raise river beds that greatly raises the
flood danger as has happened in the region of the Wenchuan earthquake since 2008
(Yang et al., 2015). This is a problem that needs to be recognized in post-earthquake
reconstruction.
Such consequences from earthquakes are long lasting.  It is estimated to have taken
three years for the Kashmir earthquake region to recover; six for the Chi-chi earthquake;
over seven for the Wenchuan earthquake and even longer for others (Saba et al., 2010;
Yang et al., 2015; Dadson et al., 2004).
There is a worry of additional landslides and rock falls after the Nepal earthquake,
especially of large ones, that might block river valleys and impound water. Indeed, both
the number and range of collapse and rock fall has clearly increased in this year's rainy
season following the earthquake. Zhou (2015) noted that before the earthquake
collapses and rockfalls only occurred on steep hill slopes on both sides of the Bo Qu
River north of Zhangmu Town, but now take place along the entire highway in this area.
He reports that the increased hazard has caused many road closures and damaged
vehicles, but no casualties as yet, because the town was evacuated after the earthquake.
The increased hazards are mainly distributed nearby along the highway between
Nyalam to Zhangmu where eighteen major landslide groups have been identified after
the earthquake by the National Disaster Reduction Center of the Ministry of Civil
Affairs using high resolution remote sensing images. Such an increase in the number of
slides should be widespread in several major valleys of southern Tibet, but relatively
few have been reported due to scarce personnel and poor transportation and
communication.

Unstable masses found to date are: the reactivated landslide group at Zhangmu,

collapse of the upper edge of the Sale Village landslide in Sale, potential failure of the
dangerous rock mass at the Rongxia Primary School, and instability of the old Natang
Village landslide and its upper edge at Chentang Town.

All of Zhangmu is located on a group of old landslides (Figs. 4a, 7a and 7b).

Discontinuous tension fissures, which are tens to hundreds of meters long, about 10 cm
wide and 2 to 4 m deep, were found at its upper edge and on its sides after the
earthquake (Figs. 8a and 8b). These fissures indicate the possibility of the failure of the
entire landslide group.

The Sale Village landslide, which resulted from the earthquake, on the slope along

the highway from Sale Village to Seqiong Village (Fig. 5b). It is nearly 600,000 $m^3$ in
volume and its fall blocked the road. Large tension fissures at its upper edge indicate a
danger of further slippage (Fig. 8c).

The dangerous rock mass at the Rongxia Village Primary School occupies a

convex portion of the cliff behind the school and appears unstable (Figs. 5c and 8d). A
rockfall occurred here during the earthquake, but the fall appears to have been
incomplete and left a cliff that lacks stability and is susceptible to further rockfall.

Natang Village near Chentang is located at the front, lower edge of an old landslide,

which is about 420 m long and 230 m wide, and consists of about 1,200,000 m$^3$ (Figs.
5d and 8e). The steep wall at its upper edge appears as two large dangerous rock blocks
which are about 60,000 m$^3$ in volume and a 1.7 m wide preexisted crack occurs between
the unstable rock blocks and the bedrock (Fig. 8f). The earthquake caused a partial
rockfall and demonstrates the dangerous instability of the mass that might come down
easily.

The danger of post-seismic debris flows also must be stressed, although these were

rare for this earthquake in the southern Tibet region; they were a serious problem in the
Wenchuan earthquake (Cui et al., 2010; Tang et al, 2011). There is, however, a
considerable amount of loose debris accumulated in mountain valleys and gullies that
could provide material for further debris flows, especially on the south slope of the
Himalaya Mountains. Rainfall, which provides excessive water to lubricate land slips
and adds weight to a lose mass, is a key factor in inducing post-seismic debris flows as
well as triggering landslides and rockfalls. There is a large difference in rainfall
between the south and north slopes of the Himalaya Mountains. The annual average
rainfall at Zhangmu on the south slope is up to 2,556.4 mm/a, whereas the annual
average rainfall in Jilong and the seat of Nyalam County on the north slope is only
880.3 mm/a and 654.0 mm/a, respectively. The rainfall on the south slope is
concentrated in the Indian Ocean summer monsoon season and induced debris flows
were already being reported in Nepal at the beginning of June. The several incised
valleys in the south mentioned above are sites of potentially dangerous post-seismic
debris flows in Tibet particularly in the three deeply incised valleys leading toward
Nepal that have a high potential for flows that could dam the rivers to form dangerous
lakes. These valleys, from west to east, are: The Gyirong Zangbo River in the upper
basin of the Trisuli River, the Boqu River and the Rongxiaqu River in the upper basin of
the Sunkoxi River (Fig. 2). Another danger spot is in the Dianchang gulley on the south
side of Zhangmu (Figs. 5a and 7a) where considerable loose debris is in a very unstable
state.

## 5. Discussion

### 5.1 Pattern of damages

The Nepal earthquake was felt over a wide region of southern Tibet. Fortunately, few casualties occurred, because of the sparse population, but there was extensive damage due to the presence of many poorly built stone and adobe buildings and the impact of landslides, collapses and rockfalls in this steep mountainous region of high relief that is similar in the Nepal region (Zekkos et al., 2017); the intensity near the Nepal border approached IX. The intensity distribution showed that the attenuation rate northward was more than twice that in either eastward or westward directions due to the absorption of energy by the major E-W-trending structure of the region and the trend of the seismic activity in the epicentral area. The intensity survey demonstrated a very wide difference in seismic performance between these poorly built buildings and well-built brick and concrete ones. In addition to the immediate damage shown by the intensity, there are the delayed effects of further dangerous land movement and an increased potential for a significant earthquake over the next several years; all of which are important in consideration of the seismic hazard in the region and post-earthquake reconstruction.

The numerous landslides, collapses and rockfalls occurred on slopes steeper than 35-45 degrees and usually at locations where previous one took place. This apparently reflects the effects of ancient earthquakes and provides new evidence for paleo-seismicity. The presence of large landslides, which either did not fail or only partially so, also suggests that larger earthquakes affected this region in the past. These sites of ancient and modern slips mark the hazardous areas in future earthquakes; an important consideration in seismic risk evaluation and the post- earthquake reconstruction process.

The Nepal earthquake both changed and brought out features that enhance the seismic hazard in the near and long term. The principal geologic dangers emanate from landslides, collapses and rockfalls in this steep terrane from ones loosened or only

partially failed immediately or new ones from ground weakened by the general ground shaking of the earthquake. These will be more common in the next three to six years or so as a delayed effect of the earthquake, especially in seasons of heavy rainfall. All of the areas of older landslips, whether or not they reactivated in this earthquake, are susceptible to reactivation and are particularly dangerous. In recent years it has been discovered that ground motion from large earthquakes results in weakened cohesiveness of the ground and causes more abundant landslips subsequently. These may clog valleys to form dangerous reservoirs. Such an increase in landslips has already been note in the study area this past summer.

Both the landslips during an earthquake and the delayed ones contribute debris to the river valleys to widened them and raise riverbeds to create conditions for flooding. This can destroy additional buildings and endanger bridges as in the area of the Wenchuan earthquake (Yang et al., 2015).

Following these characteristics, we should focus three circumstances in the assessments of seismic geological hazards (mainly refers to collapse, landslips and rockfall here) within the many strong earthquakes and high relief area:

(1) steep slopes formed by loose bodies, such as thick alluvial and residual deposits, in deep valleys;

(2) places with multiple periods of landslides, collapses and rockfalls;

(3) the revival possibility of known landslides and collapses in future earthquakes.

**5.2 Relationship between MHT and N-trending rifts**

The Nepal earthquake has likely set the stage for another forceful nearby earthquake that can be considered a delayed effect. The release of energy in a great earthquake such as the Nepal earthquake may shift the strain in the adjacent regions where other earthquakes may then occur, such as the strong earthquakes that occurred in Tibet following the Ms 8.0 Wenchuan earthquake (Wu et al, 2011). The seismic history of southern border of Tibet appears to bear this out. Large earthquakes along the south margin on the Main Frontal Thrust of the Main Himalayan Thrust are followed by

ones along the N-S-trending normal faults in the region to the north (Fig. 1). There now
is an increased concern that a significant earthquake may occur along the normal faults
in the region based on this past history

Southern Tibet is an earthquake-prone region with long E-W-trending active thrust

faults such as caused the Nepal Earthquake; less well known are the important active
normal faults and grabens just to the north (Figs. 1 and 2). These normal faults form at
least eight nearly N-S-trending rifts across southern Tibet. Geological estimates and
GPS data show that the E-W extension rates cross the rifts were 10-13 mm/a during the
Quaternary and Holocene (Armijo et al., 1986; Chen et al., 2004). Such rates are close
to the Holocene slip rate of 21±1.5 mm/yr along the Main Frontal Thrust of the Main
Himalaya Thrust (Lavé and Avouac, 2000) and to the recent GPS-based shortening rate
of 10-19 mm/yr across the Himalaya orogenic belt (Larson et al., 1999; Jouanne, et al.,
1999; Zhang et al., 2004; Bettinelli et al., 2006). There thus, appears to be a close
kinematic connection between the thrusting on the Main Himalaya Thrust and the
nearly N-S-trending normal faulting in the southern Tibet region as indicated by the
historic seismicity (Armijo et al, 1989; Molnar and Lyon-Caen, 1989).

Often within a short time interval of about one to 10 years after great earthquakes

on the Main Himalaya Thrust, strong earthquakes occur on the N-S-trending normal
faults in the southern Tibet region (Fig. 9). For example, the great Kashmir earthquake
of 1400 was followed by a M 8.0 earthquake in the Damxung-Yangbajain sector of the
northern Yadong-Gulu Rift in 1411; a M 8.1 earthquake in the western part of Nepal in
1803 was followed by a M 7.5 earthquake in the southern sector of the Cona-Oiga Rift
in 1806, and the M 7.8 Kashmir earthquake of 1905 was followed by a M 7.5
earthquake at Sangri in the northern sector of the Cona-Oiga Rift in 1915. Similarly,
after the M 8.1 1934 Nepal earthquake, a M 7.0 earthquake in the same year occurred in
the N-S-trending Gomang Co graben in northeastern Xainza County and after the 1950
M 8.6 China-Indian border earthquake, a M 7.5 earthquake occurred in 1952 in the
northern sector of the Yadong-Gulu Rift in Nagqu County.

Another delayed effect of the earthquake is the enhanced seismic hazard due to the

release in energy and the shift in strain, based on the past seismic history. The Nepal

earthquake emphasizes the close relation between the seismic activity and the dynamics in the nearly east-west stretch of deformation along the Himalaya foothills and the controlling activity along the Main Himalaya Thrust, which triggered the Nepal earthquake. Extensional forces about perpendicular to the active thrust have a history of resulting in a nearby significant normal fault earthquake following thrust movement within the subsequent 10 years that results in further destruction and fatalities. Some normal fault activity has indeed been noted in the aftershocks of the Nepal earthquake, but not nearly enough to release the expected strain.

On the first and second day after the 2015 Nepal earthquake a Mw 5.4 earthquake occurred in Nyalam County and a Ms 5.9 earthquake in Tingri County, respectively. Both are nearly N-S-trending normal faulting-type earthquakes: the former occurred in the Nyalam-Coqên Rift and the latter at the southern end of the Xainza-Dinggye Rift. However, these movements are unlikely to have released all the built up extensional force. Recently, Elliott et al. (2010) found from the InSAR and body wave seismological images of normal faulting earthquakes that the extension rate due to the contribution of the seismic energy released through normal faulting for the past 43 years in the southern Tibet region is 3-4 mm/a, which is only equivalent to 15-20% of the extension rate obtained by GPS measurements. This suggests that there still is about 80% of the energy due to extension to be released, possibly as near-future seismic activity.

The extension also may affect a set of NW-trending right-lateral strike-slip fault zones that have significant activity in the southern Tibet region. These are from west to east: The Karakorum fault zone, the Gyaring Co fault zone, and the Bengcuo fault zone (Fig. 1). Their Quaternary strike-slip rate may reach 10-20 mm/a (Armijo et al., 1989; Chevalier et al., 2005). Such faults with high strike-slip rates also can play an important role in adjusting of the nearly E-W extensional deformation in the area. For example, a M 8.0 earthquake in southwestern Nagqu in 1951, which occurred along the NW-trending Bengco fault zone, followed the 1950 M 8.6 Zayü earthquake of eastern Tibet that is known in India as Assam earthquake.

Based on past experience, the southern Tibetan region in the vicinity of the Nepal

earthquake is likely to have a normal fault earthquake within the next 10 years.
**5.3 Suggestions for regional earthquake prevention and disaster mitigation**
This investigation is preliminary and generalized, but tentative recommendations
can be issued to guide reconstruction in the region.
First, southern Tibet is a region with remarkable historical seismicity where
earthquakes and their effects cannot effectively be forecast, but a reevaluation of the
earthquake hazards should be made as soon as possible to indicate the potential dangers
noted in this survey.
Second, the relocation and reconstruction of damaged residential areas needs to
consider the potential dangers of post-seismic hazards and stability of previous
seismically induced geologic effects. Areas of ancient landslides, collapse and rockfall,
in particular need to be mapped and avoided, specially for schools, hospital, utilities
and vital government buildings and, where impossible, roads and bridges. Bridges
might be rebuilt higher in valleys where riverbeds may be raised and the flood danger
enhanced due to increased debris flow from the displaced material. And for the same
reason selection of building sites in valleys must be chosen with care. A wide selection
for new, safer sites for construction should be provided in the vast southern Tibetan
region with its very low population density.
Third, in the repair and reconstruction of buildings, new anti-seismic construction
codes must be adopted. The replacement of poorly built stone and adobe building by more
seismic resistant brick and concrete ones should be given a high priority.
Fourth, over the next 10 years there should be heightened awareness and
preparations for a possible earthquake in one of the grabens of southern Tibet.
Finally, although more detailed seismic-geological study is, of course, necessary,
the greater urgency should be directed at the construction of better anti-seismic
buildings and facilities in areas away from potential geological hazards that may be
triggered by earthquakes.

## 6. Conclusions

(1) Damages caused by Nepal earthquake in Tibet vary with the intensity, amount of rock weakened by previous movement, steepness of slope and lithology. And the damages show directional features mainly developed in the N-trending rifts in southern Tibet. However, the damages weren't related to the faulting of N-trending rifts.

(2) The earthquake induced landslide and collapse generally occurred where previous ones had taken place and correlated in size with the previous ones. Therefore, the areas of ancient landslide and collapse indicate the potential areas of danger from further landslips from torrential rains and future earthquakes.

(3) The damages directional features, paleo-earthquakes and deformational rate also suggest that the E-W extensional deformation in southern Tibet is closely associated with the Himalaya thrust fault. Then, the activity of MHT could trigger the active faulting of N-trending rifts.

## ACKNOWLEDGEMENTS

This work was supported by National Natural Science Foundation of China (No. 41571013). We would like to thank Professor Tingshan Tian and Jietang Lu of the China Institute of Geo-environment Monitoring, Professor Qiang Xu and Doctor Guang Zheng of Chengdu University of Technology, Professor Ji Duo and Baoben Xia of Geology and Mineral Resources Exploration Bureau of Xizang Autonomous Region for their participating of our field investigation. We also appreciate the help extended by the Department of Land and Resources of Xizang Autonomous Region and relevant local governments.

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

**Tables**

Table 1 Location of surveyed sites of earthquake intensity in southern Tibet

| site | coordinate | intensity | note |
|---|---|---|---|
| Lhasa city | 29.65 °N, 91.12 °E | III | felt area |
| Xaitongmoin town | 29.432 °N, 88.259 °E | III | felt area |
| Xigazê city | 29.27 °N, 88.88 °E | IV | felt area |
| Nêdong city | 29.23 °N, 91.76 °E | IV | felt area |
| Gamba town | 28.276 °N, 88.516 °E | VI | |
| Sagya town | 28.903 °N, 88.020 °E | VI | |
| Lhazê town | 29.087 °N, 87.634 °E | VI | |
| Ngamring town | 29.298 °N, 87.234 °E | VI | |
| Sangsang town | 29.420 °N, 86.724 °E | VI | |
| Saga town | 29.329 °N, 85.233 °E | VI | |
| Gyirong town | 28.856 °N, 85.297 °E | VI | |
| Tingri town | 28.661 °N, 87.122 °E | VI | |
| Dinggyê town | 28.367 °N, 87.772 °E | VI | |
| Riwu town | 28.012 °N, 87.681 °E | VI | |
| Rema villiage | 28.459 °N, 85.224 °E | VII | |
| Bangse villiage | 28.083 °N, 86.368 °E | VII | |
| Rongxia town | 28.057 °N, 86.342 °E | VII | |
| Chentang town | 27.868 °N, 87.414 °E | VII | |
| Natang villiage | 27.850 °N, 87.441 °E | VII | |
| Jilong town | 28.396 °N, 85.327 °E | VIII | |
| Sale town | 28.365 °N, 85.445 °E | VIII | |
| Guoba villiage | 28.365 °N, 85.457 °E | VIII | |
| Zuobude villiage | 28.037 °N, 86.297 °E | VIII | |
| Zhangmu town | 27.990 °N, 85.982 °E | IX | |
| Disgang villiage | 27.984 °N, 85.979 °E | IX | |
| Lixin villiage | 27.960 °N, 85.971 °E | IX | |
| Kodari town, Nepal | 27.972 °N, 85.962 °E | IX | |
| Jifu villiage | 28.374 °N, 85.329 °E | IX | |
| Chongse villiage | 28.373 °N, 85.362 °E | IX | |

Table 2 Distribution of seismic intensity in the southern Tibet region from the Nepal earthquake.

| Intensity | Area (km²) | city, county and town covered by seismic intensity | damage of building and surface |
|---|---|---|---|
| IX | 105 | The Zhangmu Town of Nyalam County, Jilong Town of Gyirong County. | Most of the mud-brick and stone piled up building were collapsed and severely damaged and some brick houses also have obvious damage and partial collapse. Collapse and landslide is widespread, and the existence of large landslides. |
| VIII | 1,945 | The Zhangmu Town and Nyalam Town of Nyalam County, Jilong Town and Sale Town of Gyirong County, Rongxia Town of Tingri County. | Some of the mud-brick and stone piled up buildings were collapsed or severely damaged, but the buildings of brick structure are mainly moderate to slightly damaged and are more of the wall cracks. Medium and small collapses and landslides are common but are rarely large landslide. |
| VII | 9,590 | Gyirong County, Nyalam County, Tingri County and Dinggye County. | A few of the mud-brick and stone piled up buildings were severely damaged, but most buildings are slightly damaged only. There are some small collapses, landslides and rockfalls along slope of valley and highway roadcuts. |
| VI | 35,460 | Zhongba County, Saga County, Gyirong County, Nyalam County, Tingri County and Dinggye County, Gamba County, Sàgya County, Ngamring County and Lhazê. | Only a few the mud-brick and stone piled up buildings were slightly damaged, and collapses and landslides are rare. A small amount of rockfall may appear near the highways roadcuts. |
| Felt area | 300,000 | Lhasa, Xigazê, Burang, Gar and Nêdong etc. | |

**Figures**

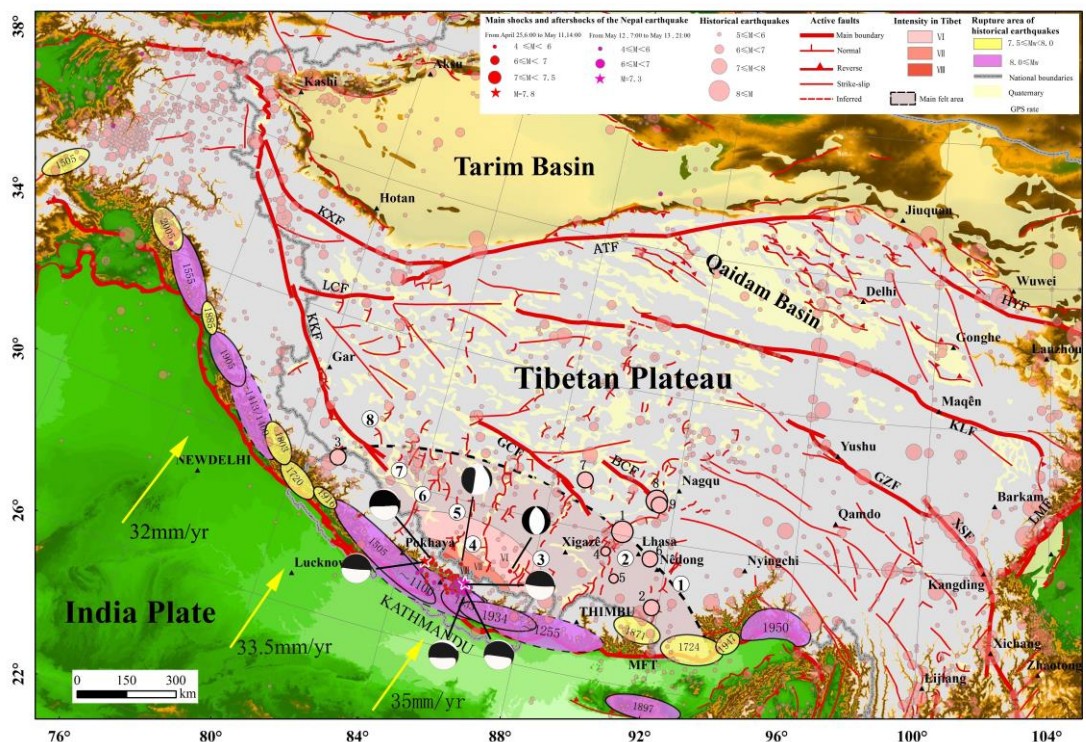

Fig.1 Principal active faults and historic earthquakes in the Himalaya Mountains, Tibetan Plateau and neighboring areas. The earthquake data is from The Science and Technology Committee and the archives in Xizang Autonomous Region, 1982; Bilham, 2004; Avouac, 2007; GPS data from Bettinelli et al, 2006; The focal mechanism solution data from USGS, 2015 a,b and Institute of Geophysics, China Earthquake Administration, 2015. Explanation: Rifts in southern Tibet, ①, Cona-Oiga rift; ②, Yadong-Gulu rift; ③, Dinggye-Xainza rift; ④, Gangga-Tangra Yumco rift; ⑤, Nyalam-Coqên rift; ⑥, Zhongba-Gêrzê rift; ⑦, Kunggyu Co-Yagra rift; ⑧, Burang-Gêgyai rift. Thrust and strike-slip faults: MFT, Main Frontal Thrust fault of Himalaya; KKF, Karakorum fault; GCF, Gyaring Co fault; BCF, Beng Co fault; GZF, Ganzi fault; XSF, Xianshuihe fault; KLF, Kunlunshan fault; LMF, Longmenshan fault; LCF, Longmu Co Fault; KXF, Kangxiwa fault; AFT, Altyn Tagh fault; HYF, Haiyuan fault. Numbers 1-9, M≥6.8 historic earthquake epicentral areas in southern Tibet: 1, 1411 M 8.0 Damxung-Yangbajain; 2, 1806 M 7.5 Cona; 3, 1883 M 7.0 Burang; 4, 1901 M 6.8 Nyêmo; 5, 1909 M 6.8 Nagarze; 6, 1915 M 7.0 Sangri; 7, 1934 M 7.0 Gomang Co of Xainza; 8, 1951 M

8.0 Beng Co of Nagqu; 9,1952 M 7.5 Gulu of Nagqu.

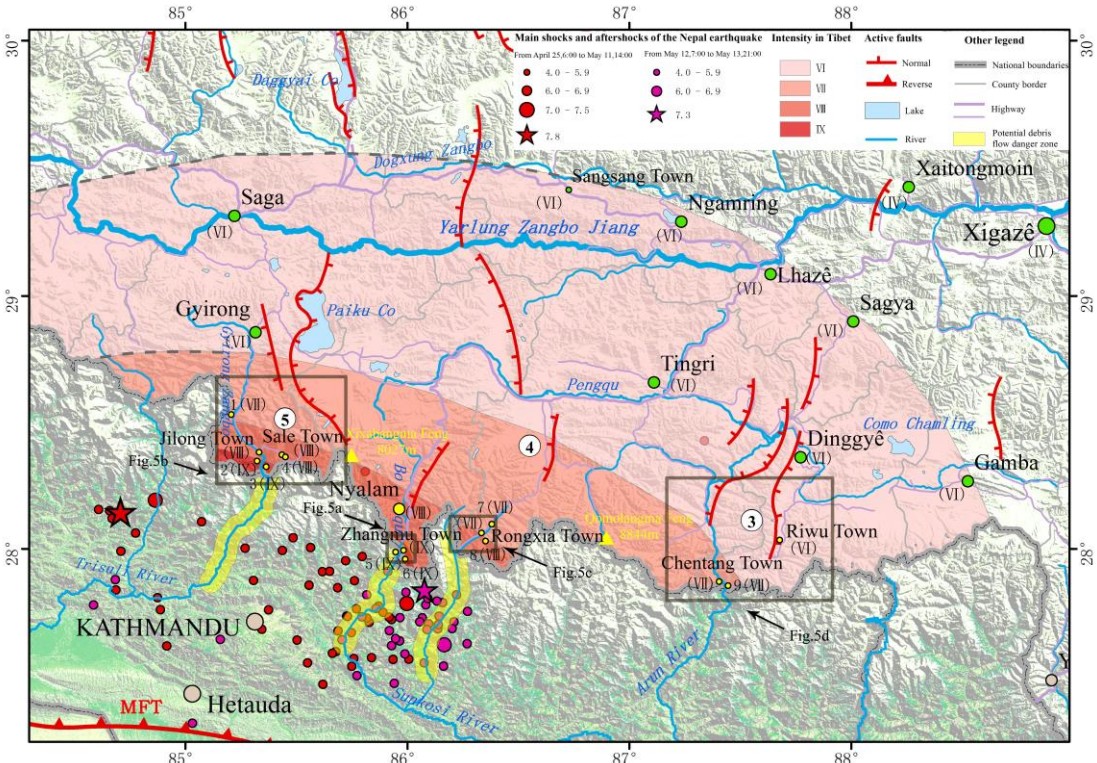

Fig. 2 Principal active faults and the distribution of seismic intensity of the 2015 Nepal earthquake in the southern Tibet region. Epicentral data from the USGS and seismic intensity from the China Earthquake Administration. The numbers and names of the principal S-N trending rifts in southern Tibet are same as on Fig. 1. The green landmarks show the sites which intensity are from China Earthquake Administration, and yellow landmarks show the spots which intensity is resulted from our field investigation. The survey spots: 1, Rema Villiage; 2, Jifu villiage; 3, Chongse Villiage; 4, Guoba Villiage; 5, Kodari Town of Nepal; 6, Lixin Viliage; 7, Bangse Villiage; 8, Zhuobude Villiage; 9, Natang Villiage.

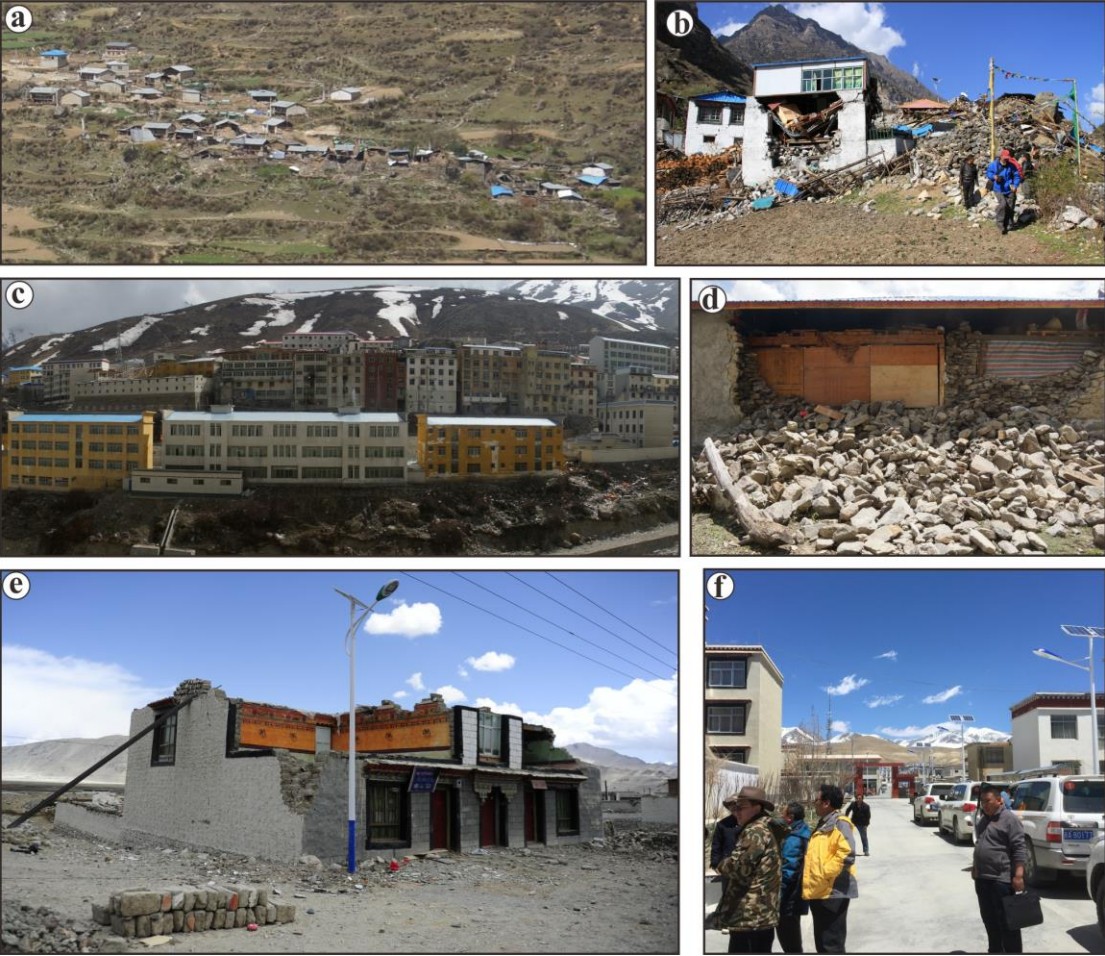

Fig. 3 The building damage resulting at different earthquake intensities: a, Many of the old stone pile or mud-brick houses collapsed, but the new brick houses rarely collapsed at Gangba Village in Sale Town in the intensity VIII zone; b, Similar building damage at Zhuobude Village of Rongxia Town in the intensity VIII zone; c, Most of buildings of brick-concrete structure did not collapse, but many walls showed obvious damage at Nyalam city located in the intensity VIII zone; d, Some walls of the stone-piled or mud-brick houses collapsed at Rema Village, Jilong County, in the intensity VII zone; e, Similar building damage at Chentang Town, Dinggy ê county, in the intensity VII zone; f, Most of the houses remain intact and only few or individual walls of buildings had apparent small cracks at Jilong county city in the intensity VI zone.

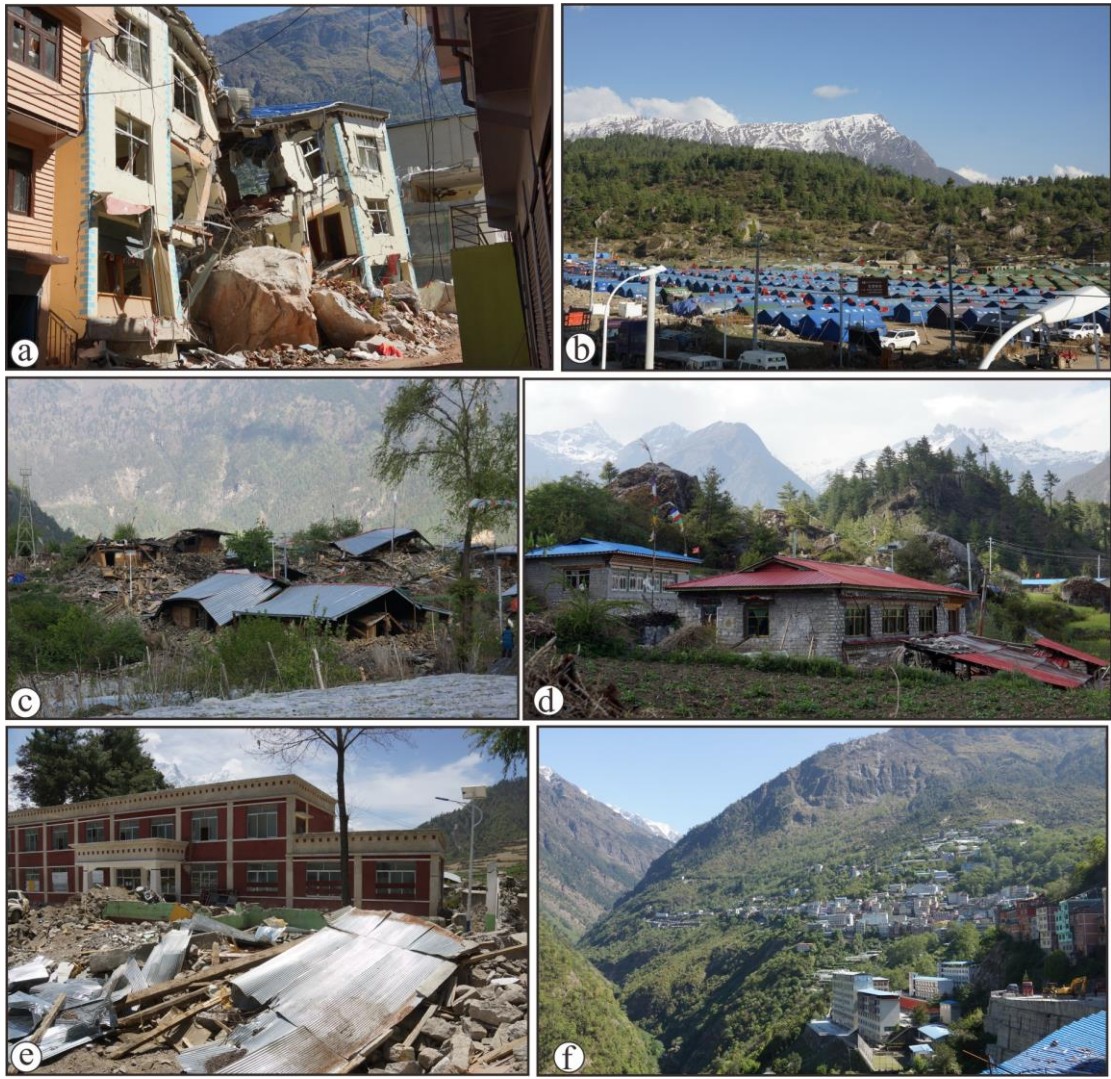

Fig. 4 Typical earthquake damage in southern Tibet and comparison of houses of different construction (locations shown in Fig. 5). Huge rockfall that smashed the resident committee office building at Disigang Village, about 0.7 km south of Zhangmu, where seven persons were killed (intensity IX) (site 1, Fig. 5a); b, A temporary settlement for earthquake survivors at Jilong; c, Destroyed houses of stone block masonry or adobe construction in Jifu Village southwest of Jilong (intensity VIII) (site 8, Fig. 5b); d, Houses of cement-bonded stone or brick construction in Jifu Village (intensity IX); e, Destroyed old houses and standing new buildings at Sale Town Primary School (intensity VIII) (site 7, Fig. 5b); f, Few collapsed houses at Zhangmu due to the brick structure or reinforced concrete construction (intensity IX).

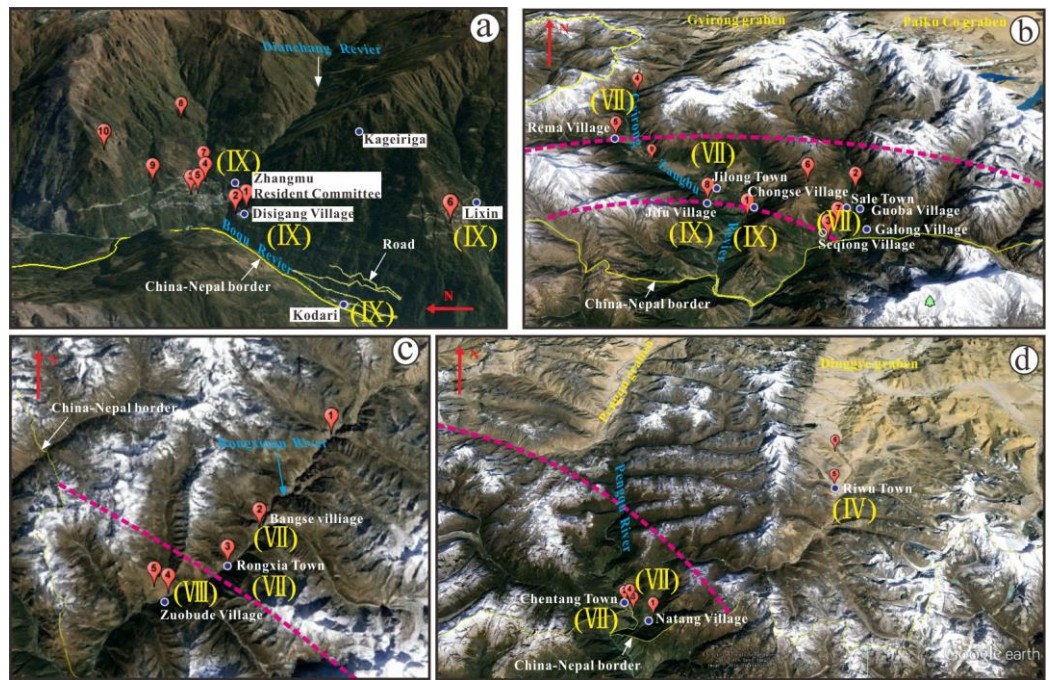

Fig. 5 Main surveyed sites of seismic effects after the Nepal earthquake, see Fig. 2 for the locations (image source Google Earth). Explanation: Roman numerals in brackets, seismic intensity values of the corresponding location; pink dotted lines, boundaries between different intensity zones. a, Zhangmu Town and vicinity; b, Jilong Town and environs; c, Rongxia Town and vicinity; d, Riwu Town to Chentang Town. Explanation: numbered balloons, sites of particular effects; red dashed lines, isoseismal boundaries.

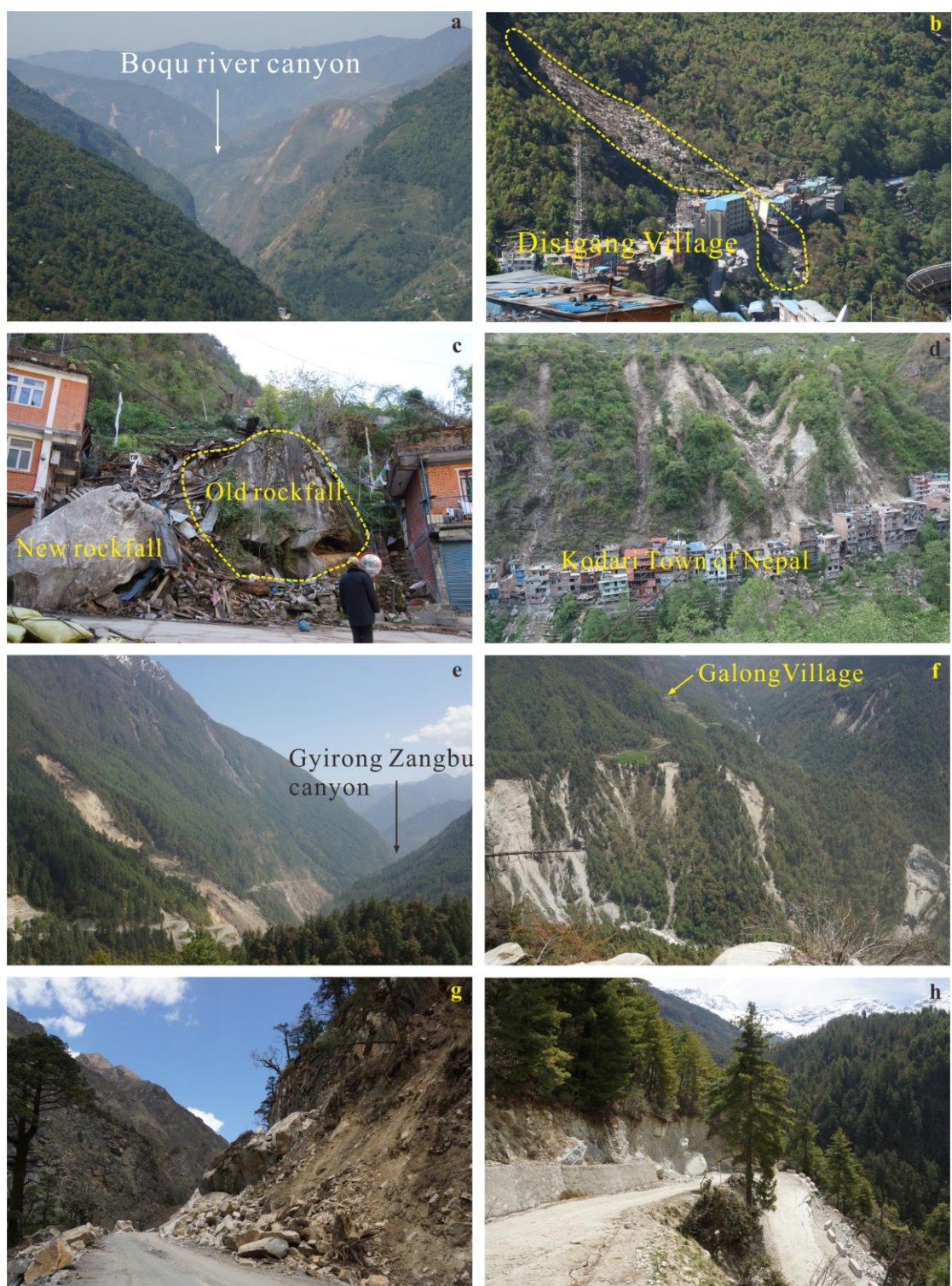

Fig. 6 Geologic effects caused during the Nepal earthquake: a, collapses in the Boqu valley; b, collapse at Disigang Village in the Boqu valley (Site1, Fig. 5a); c. new and old rockfalls at Disigang Village in the Boqu valley (Site1, Fig. 5a); d, destroyed buildings in Kodari, Nepal in the Boqu valley (Site in Fig. 4a); e, large landslide in Chongse Village in the Gyirong Zangbo valley (Site1, Fig. 5b); f, collapses in Galong Village in the Gyirong Zangbo valley (Site 7, Fig. 5b); g, collapses along highway from Gyirong County to Jilong Town in the Gyirong Zangbo

valley (Site 4, Fig. 5b); h, collapses and fissures along the highway from Jilong to ChongseVillage in the Gyirong Zangbo valley (Site1, Fig. 5b).

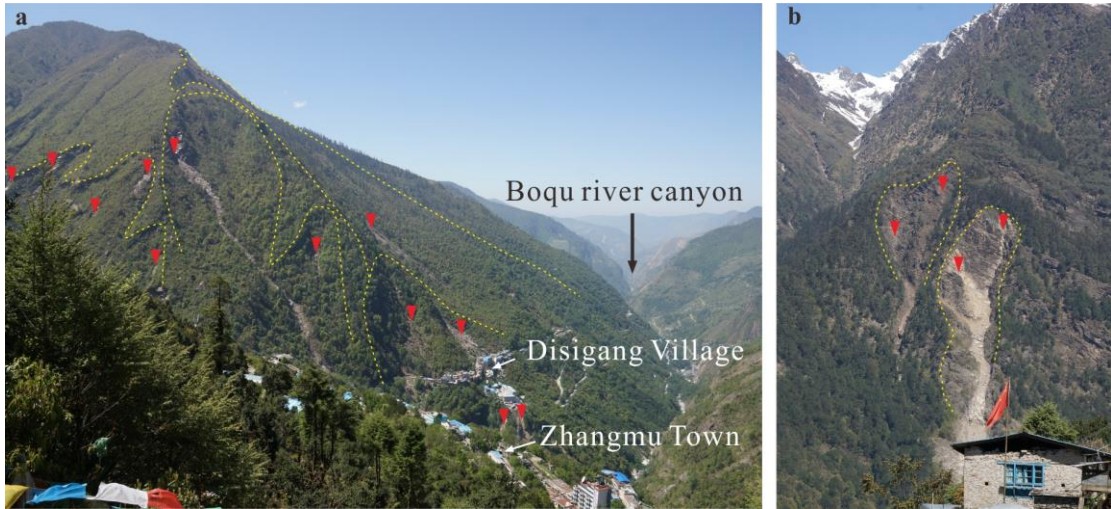

Fig. 7 New and old collapses and landslides on both banks of the Boqu River in Zhangmu Town; a. the east bank; b. the west bank. Explanation: yellow dotted line, boundary of old collapse and landslide; red triangle, new collapse during the Nepal earthquake.

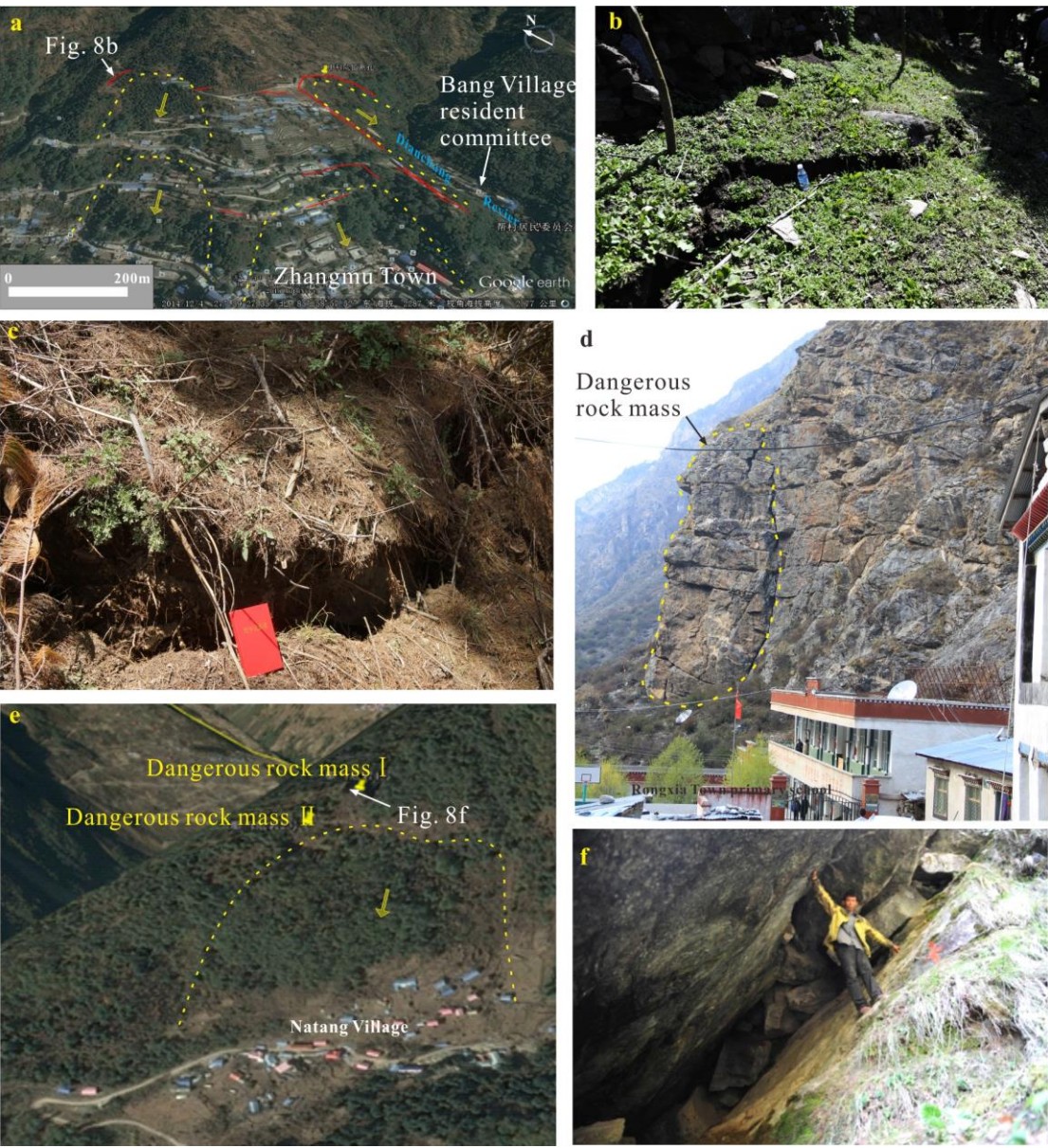

Fig. 8 Fissured and unstable rock masses formed by the earthquake that indicate hazards for additional landslides and rockfalls. Explanation: yellow dotted line, landslide group; arrow, slip direction; red line, new fissures formed during the Nepal earthquake; a, Old landslide group at Zhangmu. b. New fissure in the old landslide group at Zhangmu (site in Fig. 8a); c, Tension fissures at the back edge of Sale Village landslide (site7 in Fig. 5b); d, Dangerous rock mass at Rongxia Primary School (site 3 in Fig. 5c); e, Old landslide with unstable rock at Chentang Town (site 1 in Fig. 5d) and; f, Fissure between unstable rock and bedrock at Chentang (site in Fig. 8e).

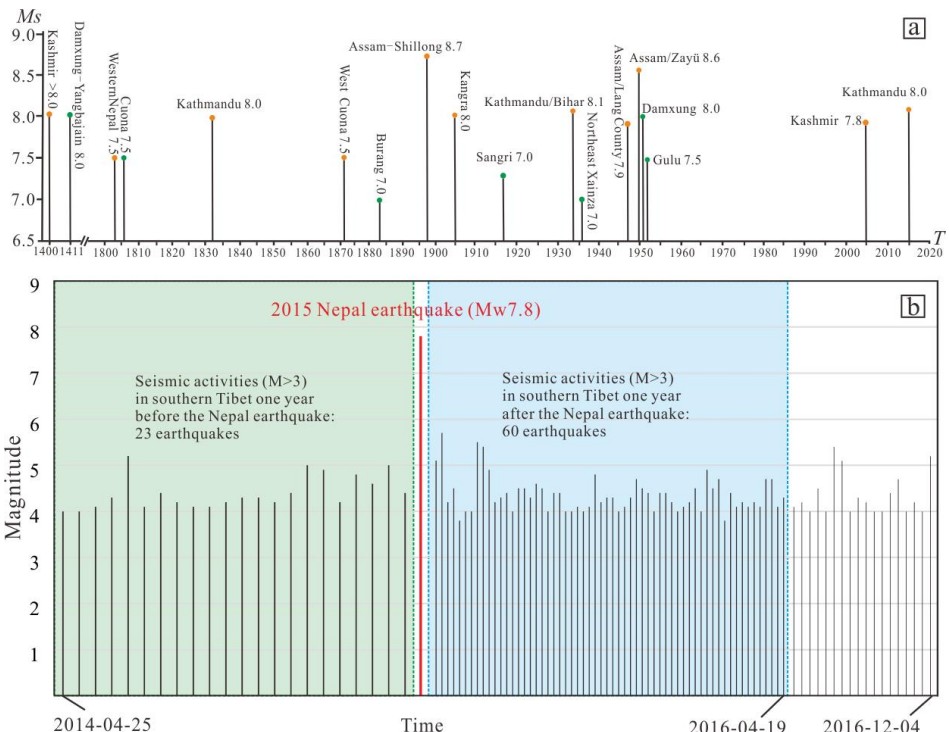

Fig.9 a: Magnitude (Ms) –Time (T) distribution of historical seismic activity along Himalaya and southern Tibet with the magnitude >7.0. The orange circles show the earthquakes occurred along Himalaya while the green circles show the earthquakes along southern Tibetan rift. b: Magnitude (M) –Time (T) distribution of seismic activity in southern Tibet in the period of one year before and after the 2015 Nepal earthquake (data came from USGS)