# Peer review of "Damages induced by the 25 April 2015 Nepal earthquake in the Tibetan border region of China and increased post-seismic hazards"

_Natural Hazards and Earth System Sciences, 2018_

## Referee Comment (RC1) · Anonymous Referee #1 · 23 Aug 2018

This manuscript presented the report of field investigation within China on the damages caused by the 2015 Nepal earthquake. Such a manuscript is a very good supplement to many published papers/reports on the earthquake caused damages in Nepal, and thus helpful for us understanding the whole seismic damages in both China and Nepal. However, the current manuscript cannot in accordance with a scientific paper be worthy published on NHESS. (1) Although I can fully understand the manuscript, the expression and writing styles is rather terrible. I suggest the manuscript should be carefully polished by a native English user. (2) The order of the figures is the manuscript is confused, for example, lines 56-65 of the introduction, where is the fig. 3. In addition, it is not suitable in the section introduction so many figures. (3) The structure of the manuscript is not good. It look like an irregular report, rather than a scientific paper.

(4) The final part, recommendations, has not valuable information (5) The manuscript focus on several aspects, such as geological setting, seismic intensity, geologic effects, and postseismic affects. However, all of the subjects are not well introduced in detailed and organized.

---

## Referee Comment (RC2) · Anonymous Referee #2 · 25 Aug 2018

The manuscript is presented as a "brief summary" (quoting) of surface effects in China associated with the 2015 Gorkha earthquake (Mw 7.8) in Nepal. The authors show a set of photographs with impressive destruction of buildings, landslides and rockfalls. Their aim is to show the severity of earthquake damage and related seismic intensity in the nearby southern Tibetan region. Moreover, they suggest a correlation with local active normal faults in the region.

Although pictures of earthquake damage are impressive and images of investigated sites in figure 4 may refer to detailed field investigations, the damage distribution in this manuscript is particularly poorly presented. The text suffers of several unclear statements and several paragraphs need to be rewritten.

My major criticism is on the authors approach on the seismic damage evaluation. Intensities are assigned throughout the manuscript but no intensity scale (MMI? MSK? EMS 98?) is presented for their damage evaluation. No information is given on how the evaluation of seismic intensity is conducted and if they used any intensity form for their damage investigations.

The correlation between the geological effects (surface damage) and normal faulting is often hard to follow. Interpretations are mixed with observations and no evidence of coseismic normal faulting that could support their inferences on surface damage distribution is presented in this so called "brief summary".

I think that this manuscript cannot be accepted for publication in NHESS.

---

## Referee Comment (RC3) · Anonymous Referee #3 · 30 Aug 2018

There are some comments:

1. A terminology Main Himalaya Thrust (MHT) is used throughout the manuscript. a confusion is that, where is that thrust? Is this the Main Central Thrust (MCT) used in Nepal or any other thrust in China? there is not clear where this thrust line MHT in the figures 1 and 2. 2. Line 153, 155 and other: the correct name of sunkoxi is Sunkoshi in Nepal. Please correct the spelling. 3. In the text, you use Armijo et al., 1988, but in the reference list you use 1986, please correct this. 4. Some of the references in the references list are not cited in the text. Please remove these from the list. Line: 345-347 Qui, P. . . ... Line: 366-369 Tang. . ..., Line : 370-372 Zhang,. . ... There are some grammatical errors in the text. Please correct them

---

## Short Comment (SC1) · 31 Aug 2018

The authors presented the results of a detail field survey of the damages in China side, caused by the 2015 Nepal Earthquake. The investigations are carried out one week after the main shock and the corresponding preliminary findings are delivered to the local government to help reducing the damages caused by secondary geohazards. As we know, strong earthquake could cause serious damages at the epicentral area, however, nearby regions were also damaged. The 2015 Nepal earthquake cause damages not only at Nepal but also in India, Pakistan, Bhutan, and the southern Tibetan region of China. There are lots of such strong earthquakes that occurred at national boarders, which need detail investigation at both countries. This study provides valuable information for us to better understand the damages caused by the 2015 Nepal earthquake.

[Figure]

Hence, I suggest this paper should be published after suitable revision.

The detail comments are listed below: 1) I suggest the author put the intensity map of Nepal side together with that of China side at Fig.2. How to incorporate the intensity maps from China side and Nepal side together? Especially whether this investigation changed the results of the intensity map from Nepal side or China side? 2) I suggest the authors provide a KMZ file as supplementary material to show the main field survey sites (if convenient, please also show the pictures of each site)

———————————————

---

## Author Comment (AC1) · 5 Sep 2018

Thank you for your detailed read on our work. The rest is the reply to your comments: (1) The Main Himalaya Thrust (MHT) is the general designation of three thrust faults along Himalaya that are Main Central Thrust (MCT), Main Boundary Thrust (MBT) and Main Front Thrust (MFT). These three faults converged into one, which is supported by seismic profile (Hauck et al., 1998; Tectonics). We will correct this in the text and figures related. (2) We will correct the spelling of Sunkoshi. (3) When we uploaded the manuscript, we mistook the rough draft to upload for discussion. In the newest manuscript, we have checked the references detailed and corrected the mistakes about references.

---

## Author Comment (AC2) · 19 Sep 2018

Thanks for your suggestions for our manuscript that are helpful to increase our paper. Based on your suggestions we have did some revisions: (1) The revised paper has been sent to a native English speaker to edit the expressions and language now. (2) We have adjusted the structure of the paper and focused on the surface damages features and seismic intensity. The supplement is the revision.

Please also note the supplement to this comment:
https://www.nat-hazards-earth-syst-sci-discuss.net/nhess-2018-195/nhess-2018-195-AC2-supplement.pdf

[Figure]

**Supplement:**

[revised manuscript text omitted]

**548 ACKNOWLEDGEMENTS**

This work was supported by grants from the Geological Survey Program of the 549 Geological Survey of China (No. 12120114002101) and the National Natural Science 550 Foundation of China (No. 41571013). We would like to thank Professor Tingshan Tian 551 and Jietang Lu of the China Institute of Geo-environment Monitoring, Professor Qiang 552 553 Xu and Doctor Guang Zheng of Chengdu University of Technology, Professor Ji Duo and Baoben Xia of Geology and Mineral Resources Exploration Bureau of Xizang 554 Autonomous Region for their participating of our field investigation. We also 555 appreciate the help extended by the Department of Land and Resources of Xizang 556 Autonomous Region and relevant local governments. 557

558

**559 **References**

Armijo R., P. Tapponnier, L. Mercier and H. Tonglin (1986). Quaternary extension in southern
Tibet :Field observation and tectonic implication, *J. Geophys. Research*,

- **562 91**(B14),13803-13872.
- Armijo R., P. Tapponnier and T. Han (1989). Late Cenozoic right-lateral strike-slip faulting in
  southern Tibet, *J. Geophys. Research*, 94, 2787-2838.
- Avouac, J-P. (2007). Dynamic processes in extensional and compressional settings mountain
   building: from earthquakes to geological deformation, *Treatise on Geophys.*, 6, 377-439.
- Bagde, M. N. and V. Petroš, (2009). Fatigue and dynamic energy behaviour of rock subjected to
  cyclical loading, *Int. J. Rock. Mech. Min.*, 46, 200–209.
- Bettinelli, P., J-P. Avouac, M. Flouzat (2006). Plate motion of India and interseismic strain in the
  Nepal Himalaya from GPS and DORIS measurements, *J. Geod.*, 80, 567–589, DOI 10.1007/s00190-006-0030-3.
- 572 Bilham, R., (2004). Earthquakes in India and the Himalaya: tectonics, geodesy and history, *Annals*573 *Geophys.*, 47(2-3), 839–858.
- 574 Chen Qizhi, J.T. Freymueller, Zhiqiang Yang, et al. (2004). Spatially variable extension in southern
  575 Tibet based on GPS, *J. Geophys. Research*, **109**, B09401, doi:10.1029/2002JB002350.
- 576 Chevalier, M.-L., F.J. Ryerson, P. Tapponnier, R.C. Finkel, J. Van Der Woerd, Haibing Li, and Qing,
  577 Liu, (2005). Slip-Rate measurements on the Karakorum fault may imply secular variations in
  578 fault Motion, *Sci.* **307**, 411-414.
- 579 China Earthquake Administration (2015). An intensity map of Tibet for the M 8.1 Nepal earthquake.
   580 http://www.cea.gov.cn/publish/dizhenj/468/553/101803/101809/20150501221123458562190/
   581 index.html.
- 582 Cui P., J-Q. Zhuang, X-Ch. Chen, et al. (2010). Characteristics and countermeasures of debris flow
  583 in Wenchuan area after the earthquake, *J. Sichuan Univ*, (*Engineer. Sci. Ed.*), 42(5), 10-19.
- Dadson, S.J., N. Hovius, H. Chen, W.B. Dade, J.C. Lin, M.L. Hsu, C.W. Lin, M.J. Horng, T.C.
  Chen, J. Milliman, C.P. Stark (2004). Earthquake-triggered increase in sediment delivery from
  an active mountain belt, *Geol.*, 32, 733–736, 2004.
- 587 Dellow, G.D. and G.T. Hancox (2006). The influence of rainfall on earthquake-induced landslides in
  588 New Zealand, in, Proceed. *Tech. Groups, Earthqs. and Urban Develop: New Zealand Geotech.*589 Soc. 2006 Sym., Nelson, New Zealand, 355–368.
- Dewey, J., R.M. Shackleton, C. Chang, and Y. Sun (1988). The tectonic evolution of the Tibetan
  Plateau, *Philos. Trans. R. Soc. London, Ser. A*, 327, 379-413.
- Elliott J.R., R.J. Walters, P.C. England, J.A. Jackson, Z. Li, and B. Parsons (2010). Extension on the
   Tibetan plateau: recent normal faulting measured by InSAR and body wave seismology,
   *Geophys. J. Internat.*, 183, 503-535. doi: 10.1111/j.1365-246X.2010.04754.x
- 595 GB/T (2008) GB/T 17742-2008, Standardization Admin. China.
- Hovius, N., P. Meunier, C.-W. Lin, H. Chen, Y.-G. Chen, S. Dadson, M.-J. Horng, M. Lines (2011).
  Prolonged seismically induced erosion and the mass balance of a large earthquake, *Earth Planet. Sc. Lett.*, 304, 347–355, doi:10.1016/j.epsl.2011.02.005.
- Hungr, O., S. Leroueil, L. Picarelli (2014) The Varnes classification of landslide types, an update,
   *Landslides*, 11, 167–194, doi:10.1007/s10346-013-0436-y.
- Institute of Geophysics, (2015). The M5.9 Tingri earthquake of April 25 2015 in Tibet, *China Earthq. Admin.*, http://www.cea-igp.ac.cn/tpxw/272116.shtml.
- IRIS (2015). Special event: Nepal, Incorp. Research Insts for Seis,
   http://ds.iris.edu/ds/nodes/dmc/specialevents/2015/04/25/nepal.
- Jouanne, F., J.L. Mugnier, M.R. Pandey, J.F. Gamond, P. LeFort, L. Serrurier, C. Vigny, J.P. Avouac
  and the Idylhim members (1999). Oblique convergence in the Himalayas of western Nepal
  deduced from preliminary results of GPS measurements, *Geophys. Research Letts.* 26,
  1933–1936.
- Larson, K., R. Burgmann, R. Bilham, J.T. Freymueller (1999). Kinematics of the India-Eurasia
  collision zone from GPS measurements, *J. Geophys. Res.*, **104**, 1077–93.
- 611 Lave , J., and J.P. Avouac (2000). Active folding of fluvial terraces across the Siwaliks Hills,

- 612 Himalayas of central Nepal, J. Geophys. Research, **105**, 5735–5770.
- Li, G., K.H.R Moelle, J.A Lewis (1992). Fatigue crack growth in brittle sandstones, *Int. J. Rock. Mech. Min.*, 29, 469–477.
- Massey, C.I., F. Della Pasqua, T. Taig, B. Lukovic, W. Ries, D. Heron, G. Archibald (2014a)
  Canterbury Earthquakes 2010/11Port Hills Slope Stability: Risk assessment for Redcliffs, *GNS Sci., Wellington, New Zealand*, p. 123 C Appendices.
- Massey, C.I., T. Taig, F. Della Pasqua, B. Lukovic, W. Ries, G.Archibald (2014b). Canterbury
  Earthquakes 2010/11 Port Hills Slope Stability: Debris avalanche risk assessment for
  Richmond Hill, *GNS Sci. Consultancy Rept.* 2014/34.
- Molnar, P., and H. Lyon-Caen (1989). Fault plane solutions of earthquakes and active tectonics of
  the Tibetan Plateau and its margins, *Geophys. J. Internat.*, 99, 123–153.
- Nara, Y., K. Morimoto, T. Yoneda, N. Hiroyoshi, K. Kaneko (2011) Effects of humidity and
  temperature on subcritical crack growth in sandstone, Int. J. Solids Structures, 48, 1130–1140.
- 625 Parker, R.N., G.T. Hancox, D.N. Petley, Massey, A.L. Densmore, N. J. Rosser (2015). Spatial distributions of earthquake-induced landslides and hillslope Preconditioning in the northwest 626 Island, New Zealand, Earth Surface Dynamics, 3, (4): 501 627 South DOI: 628 10.5194/esurf-3-501-2015
- Petley, D.N., S.A. Dunning, N.J. Rosser (2005) The analysis of global landslide risk through the
  creation of a database of worldwide landslide fatalities, in: Landslide Risk Management, eds.
  Hungr, O., R. Fell, R. Couture, E. Eberhardt, Balkema, The Netherlands.
- Saba, S. B., M. van der Meijde, H. van der Werff (2010). Spatiotemporal landslide detection for the
  2005 Kashmir earthquake region, Geomorph. 124, 17–25,
  doi:10.1016/j.geomorph.2010.07.026, 2010.
- Tang, Ch., W-L. Li and J. Ding (2011). Field investigation and research on giant debris flow on
  august 14, 2010 in Yingxiu town, epicenter of Wenchuan earthquake, *Earth Sci.-J. China Univ. Geosci.*, 36(1), 172-180. doi:10.3799/dqkx.2011.018
- The Science and Technology Committee and the archives in Xizang Autonomous Region (1982).
  Tibet earthquakes; historical compilation, (v.1), *People's Publishing House, Xizang*, 1-583 (in
  Chinese).
- 641 USGS (2015a). Updated finite fault results for the Apr 25, 2015 Mw 7.9 35 km E of Lamjung, Nepal
  642 Earthquake (Version 2), U.S. Geol. Sur., Nat'l. Earthq. Info. Center,
  643 http://earthquake.usgs.gov/earthquakes/eventpage/us20002926#scientific finite fault
- 644 USGS (2015b). Updated finite fault results for the May 12, 2015 Mw 7.3 22 km SE of Zham, China
  645 Earthquake (Version 2), U.S. Geol. Sur. Nat'l Earthq. Info. Center,
  646 http://earthquake.usgs.gov/earthquakes/eventpage/us20002ejl#scientific\_finitefault
- Wang, Xiuying, Zhenlin Han (2010). Modeling of landslides hazards induced by the 2008
  Wenchuan earthquake using ground motion parameters, in Xie, editor, Rock stress and earthquakes, *Taylor & Francis Group*, London, p. 297-304, ISBN\_978.0.415.60165.8
- Wu, Zhenhan., P.J. Barosh, Zhonghai Wu, Daogong Hu, Xun Zhan and Peisheng Ye. (2008), Vast
  early Miocene lakes of the central Tibetan Plateau, *Geol. Soc. Amer., Bull.*, 120, 1326-1337.
- Wu, Zhonghai, Peisheng Ye, P.J. Barosh and Zhenhan Wu (2011). The October 6, 2008 Mw 6.3
- 653 magnitude Damxung earthquake, Yadong-Gulu rift, Tibet, and implications for present-day
- crustal deformation within Tibet, *J. Asian Earth Sci.*, **40**, (4), 943–957.
- Zhang, P.Z., Z. Shen, M. Wang, W.J. Gan, R. Burgmann and P. Molnar (2004). Continuous
  deformation of the Tibetan Plateau from global positioning system data, *Geol.*, 32, 809-812.
- Zhou, Chengcan (2015). Personal communication, November, 2015. *Tibetan Environmental Monitoring Station*.
- 459 Yang, W.T., M. Wang, N. Kerle, C.J. Van Westen, L.Y. Liu, and P.J. Shi (2015). Analysis of changes

[revised manuscript text omitted]

**Figures**

---

## Author Response (AR1)

Dear editor and reviewers,

Thanks for your careful consideration on our manuscript. Based on the suggestions, we made corresponding modification on this article.

**RC1**

Thanks for your suggestions for our manuscript that are helpful to increase our paper. Based on your suggestions we have did some revisions:

(1) The revised paper has been sent to a native English speaker to edit the expressions and language.

(2) We adjusted the structure of this manuscript. In the newest version, we added the chapter "Discussion" that focused on the spatial and temporal features of surface damage after Nepal earthquake and the relationship between the north trending rifts and main Himalaya thrust fault.

**RC2**

(1) The name of the intensity scale used in China is "the Chinese seismic intensity scale (GB/T 17742-2008)" which is a new revised national standard of seismic intensity in 2008. The intensity scale adopted the system of 12 degrees, and is revised from the earliest Chinese seismic intensity scale with reference to the Soviet Union Medvedev intensity which is the intensity vary from the Mercalli-Cancani intensity scale. The feeling area of the earthquake have been marked in figure 1 roughly, due to the vast and sparsely populated in Tibet, therefore, a more detailed scope is difficult to be accurately defined.

(2) Although the specific relationship is not demonstrated clearly, the close geodynamic relationship between E-W deformation in southern Tibet and MHT is supported by seismic activity, deformation rate and many other evidence. For example, two Ms  $\sim$ 5.0 earthquake occurred along N-trending rifts, which was triggered by 2015 Nepal earthquake along the MHT. And, this seismic regular pattern also presented in paleo-earthquakes.

**RC03**

(1) The Main Himalaya Thrust (MHT) is the general designation of three thrust faults along Himalaya that are Main Central Thrust (MCT), Main Boundary Thrust (MBT) and Main Front Thrust (MFT). These three faults converged into one, which is supported by seismic profile (Hauck et al., 1998; Tectonics). We will correct this in the text and figures related.

(2) We will correct the spelling of Sunkoshi.

(3) When we uploaded the manuscript, we mistook the rough draft to upload for discussion. In the newest manuscript, we have checked the references detailed and corrected the mistakes about references.

Thank you for your suggestions for our manuscript that are helpful to increase our paper. We will seriously consider your suggestions and make corresponding modifications.

(1) It is difficult to integrate the intensity map of Tibet and Nepal for that the evaluation scale may be different. And we didn't survey in Nepal. Therefore, we didn't put the intensity map of Nepal side together with that of China side. However, we will try our best to do this work based on many other published materials next.

(2) We have make the KMZ file and part of corresponding photos to show the surveyed sites, which would be as supplement.

Sincerely, Zhonghai Wu

**Effects of the 25 April 2015 Nepal earthquake in the Tibetan**

border region of China and increased post-seismic hazards

Zhonghai Wu $^{a^{\ast}}$  Patrick J. Barosh $^{b\ast},~Xin$  Yao $^{a},$  Yongqiang Xu $^{c},$  Guanghao Ha $^{a}$  and Jie Liu $^{d}$

[revised manuscript text omitted]

**ACKNOWLEDGEMENTS**

This work was supported by grants from the Geological Survey Program of the Geological Survey of China (No. 12120114002101) and the National Natural Science Foundation of China (No. 41571013). We would like to thank Professor Tingshan Tian and Jietang Lu of the China Institute of Geo-environment Monitoring, Professor Qiang Xu and Doctor Guang Zheng of Chengdu University of Technology, Professor Ji Duo and Baoben Xia of Geology and Mineral Resources Exploration Bureau of Xizang Autonomous Region for their participating of our field investigation. We also appreciate the help extended by the Department of Land and Resources of Xizang Autonomous Region and relevant local governments.

[revised manuscript text omitted]

**Figures**

---

## Referee Report (RR1)

[referee-annotated manuscript omitted]

---

## Author Response (AR2)

Dear editor and reviewers,

Thanks for your careful consideration on our manuscript. Based on the suggestions, we made corresponding modification on this article. The following is the revision description of the paper and the reply to the comments of referee review 2.

(1) In this manuscript, we presented the surface damage features in Tibet after the 2015 Nepal earthquake based on the field survey. The surveyed sites and damage distribution now are showed on the Google earth map (a KML file) in the supporting information. And, it is also exhibited by a table in the manuscript (Table 1).

(2) The manuscript has been reorganized about the structure. In the newest version, we added the chapter "methods and data", "results" and "discussion" to well-organized present the surface damage features and pattern of damage.

(3) The name of the intensity scale used in China is "the Chinese seismic intensity scale (GB/T 17742-2008)" which is a new revised national standard of seismic intensity in 2008. The intensity scale adopted the system of 12 degrees, and is revised from the earliest Chinese seismic intensity scale with reference to the Soviet Union Medvedev intensity which is the intensity vary from the Mercalli-Cancani intensity scale. The feeling area of the earthquake have been marked in figure 1 roughly, due to the vast and sparsely populated in Tibet, therefore, a more detailed scope is difficult to be accurately defined. In the supporting files, we supplied the "Seismic Intensity Table of China" in the "nhess-2018-195-supplement-version2".

(4) The evaluate on of seismic intensity was mainly conducted by field investigation including the consultant to the people in the damaged area. And, the investigation included the evaluation of the damage degree of buildings. Also, remote sense interpretation in some of local sites were conducted. Besides, this seismic intensity in the study were also benefited from the results given by China Earthquake Administration after the earthquake.

(5) Although the specific relationship is not demonstrated clearly, the close geodynamic relationship between E-W deformation in southern Tibet and MHT is supported by seismic activity, deformation rate and many other evidence. For example, two Ms ~5.0 earthquake occurred along N-trending rifts, which was triggered by 2015 Nepal earthquake along the MHT. And, this seismic regular pattern also presented in paleo-earthquakes.

Sincerely,

Zhonghai Wu